# Evidential Deep Learning to Quantify Classification Uncertainty

**Murat Sensoy**
Department of Computer Science
Ozyegin University, Turkey
murat.sensoy@ozyegin.edu.tr

**Lance Kaplan**
US Army Research Lab
Adelphi, MD 20783, USA
lkaplan@ieee.org

**Melih Kandemir**
Bosch Center for Artificial Intelligence
Robert-Bosch-Campus 1, 71272 Renningen, Germany
melih.kandemir@bosch.com

## Abstract

Deterministic neural nets have been shown to learn effective predictors on a wide range of machine learning problems. However, as the standard approach is to train the network to minimize a prediction loss, the resultant model remains ignorant to its prediction confidence. Orthogonally to Bayesian neural nets that indirectly infer prediction uncertainty through weight uncertainties, we propose explicit modeling of the same using the theory of subjective logic. By placing a Dirichlet distribution on the class probabilities, we treat predictions of a neural net as subjective opinions and learn the function that collects the evidence leading to these opinions by a deterministic neural net from data. The resultant predictor for a multi-class classification problem is another Dirichlet distribution whose parameters are set by the continuous output of a neural net. We provide a preliminary analysis on how the peculiarities of our new loss function drive improved uncertainty estimation. We observe that our method achieves unprecedented success on detection of out-of-distribution queries and endurance against adversarial perturbations.

## 1   Introduction

The present decade has commenced with the deep learning approach shaking the machine learning world [20]. New-age deep neural net constructions have exhibited amazing success on nearly all applications of machine learning thanks to recent inventions such as dropout [30], batch normalization [13], and skip connections [11]. Further ramifications that adapt neural nets to particular applications have brought unprecedented prediction accuracies, which in certain cases exceed human-level performance [5, 4]. While one side of the coin is a boost of interest and investment on deep learning research, the other is an emergent need for its robustness, sample efficiency, security, and interpretability.

On setups where abundant labeled data are available, the capability to achieve sufficiently high accuracy by following a short list of rules of thumb has been taken for granted. The major challenges of the upcoming era, hence, are likely to lie elsewhere rather than test set accuracy improvement. For instance, is the neural net able to identify data points belonging to an unrelated data distribution? Can it simply say *"I do not know"* if we feed in a cat picture after training the net on a set of handwritten digits? Even more critically, can the net protect its users against adversarial attacks? These questions have been addressed by a stream of research on Bayesian Neural Nets (BNNs) [8, 18, 26], which estimate prediction uncertainty by approximating the moments of the posterior predictive distribution.

This holistic approach seeks for a solution with a wide set of practical uses besides uncertainty estimation, such as automated model selection and enhanced immunity to overfitting.

In this paper, we put our full focus on the uncertainty estimation problem and approach it from a *Theory of Evidence* perspective [7, 14]. We interpret *softmax*, the standard output of a classification network, as the parameter set of a categorical distribution. By replacing this parameter set with the parameters of a Dirichlet density, we represent the predictions of the learner as a distribution over possible softmax outputs, rather than the point estimate of a softmax output. In other words, this density can intuitively be understood as a *factory* of these point estimates. The resultant model has a specific loss function, which is minimized subject to neural net weights using standard backprop.

In a set of experiments, we demonstrate that this technique outperforms state-of-the-art BNNs by a large margin on two applications where high-quality uncertainty modeling is of critical importance. Specifically, the predictive distribution of our model approaches the maximum entropy setting much closer than BNNs when fed with an input coming from a distribution different from that of the training samples. Figure 1 illustrates how sensibly our method reacts to the rotation of the input digits. As it is not trained to handle rotational invariance, it sharply reduces classification probabilities and increases the prediction uncertainty after circa $50°$ input rotation. The standard softmax keeps reporting high confidence for incorrect classes for high rotations. Lastly, we observe that our model is clearly more robust to adversarial attacks on two different benchmark data sets.

All vectors in this paper are column vectors and are represented in bold face such as $\mathbf{x}$ where the $k$-th element is denoted as $x_k$. We use $\odot$ to refer to the Hadamard (element-wise) product.

## 2 Deficiencies of Modeling Class Probabilities with Softmax

The gold standard for deep neural nets is to use the softmax operator to convert the continuous activations of the output layer to class probabilities. The eventual model can be interpreted as a multinomial distribution whose parameters, hence discrete class probabilities, are determined by neural net outputs. For a $K-$class classification problem, the likelihood function for an observed tuple $(\mathbf{x}, \mathbf{y})$ is

$$Pr(y|\mathbf{x}, \theta) = \mathrm{Mult}(y|\sigma(f_1(\mathbf{x}, \theta)), \cdots, \sigma(f_K(\mathbf{x}, \theta))),$$

where $\mathrm{Mult}(\cdots)$ is a multinomial mass function, $f_j(\mathbf{x}, \theta)$ is the $j$th output channel of an arbitrary neural net $f(\cdot)$ parametrized by $\theta$, and $\sigma(u_j) = e^{u_j} / \sum_{i=1}^{K} e^{u_K}$ is the softmax function. While the continuous neural net is responsible for adjusting the ratio of class probabilities, softmax squashes these ratios into a simplex. The eventual softmax-squashed multinomial likelihood is then maximized with respect to the neural net parameters $\theta$. The equivalent problem of minimizing the negative log-likelihood is preferred for computational convenience

$$-\log p(y|\mathbf{x}, \theta) = -\log \sigma(f_y(\mathbf{x}, \theta))$$

which is widely known as *the cross-entropy loss*. It is noteworthy that the probabilistic interpretation of the cross-entropy loss is mere Maximum Likelihood Estimation (MLE). As being a frequentist technique, MLE is not capable of inferring the predictive distribution variance. Softmax is also notorious with inflating the probability of the predicted class as a result of the exponent employed on the neural net outputs. The consequence is then unreliable uncertainty estimations, as the distance of the predicted label of a newly seen observation is not useful for the conclusion besides its comparative value against other classes.

Inspired from [9] and [24], on the left side of Figure 1, we demonstrate how the LeNet [22] fails to classify an image of digit 1 from MNIST dataset when it is continuously rotated in the counterclockwise direction. Commonly to many standardized architectures, LeNet estimates classification probabilities with the softmax function. As the image is rotated it fails to classify the image correctly; the image is classified as 2 or 5 based on the degree of rotation. For instance, for small degrees of rotation, the image is correctly classified as 1 with high probability values. However, when the image is rotated between $60 − 100$ degrees, it is classified as 2. The network starts to classify the image as 5 when it is rotated between $110 − 130$ degrees. While the classification probability computed using the softmax function is quite high for the misclassified samples (see Figure 1, left panel), our approach proposed in this paper can accurately quantify uncertainty of its predictions (see Figure 1, right panel).

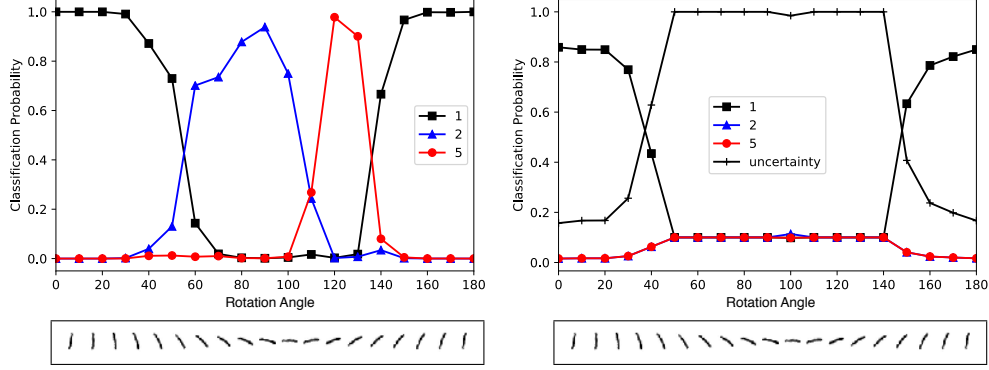

Figure 1: Classification of the rotated digit 1 (at bottom) at different angles between 0 and 180 degrees. **Left:** The classification probability is calculated using the *softmax* function. **Right:** The classification probability and uncertainty are calculated using the proposed method.

## 3 Uncertainty and the Theory of Evidence

The Dempster–Shafer Theory of Evidence (DST) is a generalization of the Bayesian theory to subjective probabilities [7]. It assigns belief masses to subsets of a frame of discernment, which denotes the set of exclusive possible states, e.g., possible class labels for a sample. A belief mass can be assigned to any subset of the frame, including the whole frame itself, which represents the belief that the truth can be any of the possible states, e.g., any class label is equally likely. In other words, by assigning all belief masses to the whole frame, one expresses *'I do not know'* as an opinion for the truth over possible states [14]. Subjective Logic (SL) formalizes DST's notion of belief assignments over a frame of discernment as a Dirichlet Distribution [14]. Hence, it allows one to use the principles of evidential theory to quantify belief masses and *uncertainty* through a well-defined theoretical framework. More specifically, SL considers a frame of $K$ mutually exclusive singletons (e.g., class labels) by providing a belief mass $b_k$ for each singleton $k = 1, \ldots, K$ and providing an overall uncertainty mass of $u$. These $K + 1$ mass values are all non-negative and sum up to one, i.e.,

$$u + \sum_{k=1}^{K} b_k = 1,$$

where $u \geq 0$ and $b_k \geq 0$ for $k = 1, \ldots, K$. A belief mass $b_k$ for a singleton $k$ is computed using the evidence for the singleton. Let $e_k \geq 0$ be the evidence derived for the $k^{th}$ singleton, then the belief $b_k$ and the uncertainty $u$ are computed as

$$b_k = \frac{e_k}{S} \quad \text{and} \quad u = \frac{K}{S}, \tag{1}$$

where $S = \sum_{i=1}^{K} (e_i + 1)$. Note that the uncertainty is inversely proportional to the total evidence. When there is no evidence, the belief for each singleton is zero and the uncertainty is one. Differently from the Bayesian modeling nomenclature, we term *evidence* as a measure of the amount of support collected from data in favor of a sample to be classified into a certain class. A belief mass assignment, i.e., subjective opinion, corresponds to a Dirichlet distribution with parameters $\alpha_k = e_k + 1$. That is, a subjective opinion can be derived easily from the parameters of the corresponding Dirichlet distribution using $b_k = (\alpha_k - 1)/S$, where $S = \sum_{i=1}^{K} \alpha_i$ is referred to as the Dirichlet strength.

The output of a standard neural network classifier is a probability assignment over the possible classes for each sample. However, a Dirichlet distribution parametrized over evidence represents the density of each such probability assignment; hence it models second-order probabilities and uncertainty [14].

The Dirichlet distribution is a probability density function (pdf) for possible values of the probability mass function (pmf) $\mathbf{p}$. It is characterized by $K$ parameters $\boldsymbol{\alpha} = [\alpha_1, \cdots, \alpha_K]$ and is given by

$$D(\mathbf{p}|\boldsymbol{\alpha}) = \begin{cases} \frac{1}{B(\boldsymbol{\alpha})} \prod_{i=1}^{K} p_i^{\alpha_i - 1} & \text{for } \mathbf{p} \in \mathcal{S}_K, \\ 0 & \text{otherwise,} \end{cases}$$

where $\mathcal{S}_K$ is the $K$-dimensional unit simplex,

$$\mathcal{S}_K = \left\{ \mathbf{p} \Big| \sum_{i=1}^{K} p_i = 1 \text{ and } 0 \le p_1, \dots, p_K \le 1 \right\}$$

and $B(\boldsymbol{\alpha})$ is the $K$-dimensional multinomial beta function [19].

Let us assume that we have $\mathbf{b} = \langle 0, \dots, 0 \rangle$ as belief mass assignment for a 10-class classification problem. Then, the prior distribution for the classification of the image becomes a uniform distribution, i.e., $D(\mathbf{p}|\langle 1, \dots, 1 \rangle)$ — a Dirichlet distribution whose parameters are all ones. There is no observed evidence, since the belief masses are all zero. This means that the opinion corresponds to the uniform distribution, does not contain any information, and implies total uncertainty, i.e., $u = 1$. Let the belief masses become $\mathbf{b} = \langle 0.8, 0, \dots, 0 \rangle$ after some training. This means that the total belief in the opinion is 0.8 and remaining 0.2 is the uncertainty. Dirichlet strength is calculated as $S = 10/0.2 = 50$, since $K = 10$. Hence, the amount of new evidence derived for the first class is computed as $50 \times 0.8 = 40$. In this case, the opinion would correspond to the Dirichlet distribution $D(\mathbf{p}|\langle 41, 1, \dots, 1 \rangle)$.

Given an opinion, the expected probability for the $k^{th}$ singleton is the mean of the corresponding Dirichlet distribution and computed as

$$\hat{p}_k = \frac{\alpha_k}{S}. \tag{2}$$

When an observation about a sample relates it to one of the $K$ attributes, the corresponding Dirichlet parameter is incremented to update the Dirichlet distribution with the new observation. For instance, detection of a specific pattern on an image may contribute to its classification into a specific class. In this case, the Dirichlet parameter corresponding to this class should be incremented. This implies that the parameters of a Dirichlet distribution for the classification of a sample may account for the evidence for each class.

In this paper, we argue that a neural network is capable of forming opinions for classification tasks as Dirichlet distributions. Let us assume that $\boldsymbol{\alpha_i} = \langle \alpha_{i1}, \dots, \alpha_{iK} \rangle$ is the parameters of a Dirichlet distribution for the classification of a sample $i$, then $(\alpha_{ij} - 1)$ is the total evidence estimated by the network for the assignment of the sample $i$ to the $j^{th}$ class. Furthermore, given these parameters, the epistemic uncertainty of the classification can easily be computed using Equation 1.

## 4  Learning to Form Opinions

The softmax function provides a point estimate for the class probabilities of a sample and does not provide the associated uncertainty. On the other hand, multinomial opinions or equivalently Dirichlet distributions can be used to model a probability distribution for the class probabilities. Therefore, in this paper, we design and train neural networks to form their multinomial opinions for the classification of a given sample $i$ as a Dirichlet distribution $D(\mathbf{p}_i|\boldsymbol{\alpha}_i)$, where $\mathbf{p}_i$ is a simplex representing class assignment probabilities.

Our neural networks for classification are very similar to classical neural networks. The only difference is that the *softmax* layer is replaced with an activation layer, e.g., *ReLU*, to ascertain non-negative output, which is taken as the evidence vector for the predicted Dirichlet distribution.

Given a sample $i$, let $f(\mathbf{x}_i|\Theta)$ represent the evidence vector predicted by the network for the classification, where $\Theta$ is network parameters. Then, the corresponding Dirichlet distribution has parameters $\boldsymbol{\alpha}_i = f(\mathbf{x}_i|\Theta) + 1$. Once the parameters of this distribution is calculated, its mean, i.e., $\boldsymbol{\alpha}_i/S_i$, can be taken as an estimate of the class probabilities.

Let $\mathbf{y}_i$ be a one-hot vector encoding the ground-truth class of observation $\mathbf{x}_i$ with $y_{ij} = 1$ and $y_{ik} = 0$ for all $k \ne j$, and $\boldsymbol{\alpha}_i$ be the parameters of the Dirichlet density on the predictors. First, we can treat $D(\mathbf{p}_i|\boldsymbol{\alpha}_i)$ as a prior on the likelihood $\text{Mult}(\mathbf{y}_i|\mathbf{p}_i)$ and obtain the negated logarithm of the marginal likelihood by integrating out the class probabilities

$$\mathcal{L}_i(\Theta) = -\log\left( \int \prod_{j=1}^{K} p_{ij}^{y_{ij}} \frac{1}{B(\boldsymbol{\alpha}_i)} \prod_{j=1}^{K} p_{ij}^{\alpha_{ij}-1} d\mathbf{p}_i \right) = \sum_{j=1}^{K} y_{ij} \Big( \log(S_i) - \log(\alpha_{ij}) \Big) \tag{3}$$

and minimize with respect to the $\boldsymbol{\alpha}_i$ parameters. This technique is well-known as the Type II Maximum Likelihood.

Alternatively, we can define a loss function and compute its Bayes risk with respect to the class predictor. Note that while the above loss in Equation 3 corresponds to the Bayes classifier in the PAC-learning nomenclature, ones we will present below are Gibbs classifiers. For the cross-entropy loss, the Bayes risk will read

$$\mathcal{L}_i(\Theta) = \int \left[ \sum_{j=1}^K -y_{ij} \log(p_{ij}) \right] \frac{1}{B(\boldsymbol{\alpha}_i)} \prod_{j=1}^K p_{ij}^{\alpha_{ij}-1} d\mathbf{p}_i = \sum_{j=1}^K y_{ij} \Big( \psi(S_i) - \psi(\alpha_{ij}) \Big), \quad (4)$$

where $\psi(\cdot)$ is the *digamma* function. The same approach can be applied also to the sum of squares loss $||\mathbf{y}_i - \mathbf{p}_i||_2^2$, resulting in

$$\mathcal{L}_i(\Theta) = \int ||\mathbf{y}_i - \mathbf{p}_i||_2^2 \frac{1}{B(\boldsymbol{\alpha}_i)} \prod_{i=1}^K p_{ij}^{\alpha_{ij}-1} d\mathbf{p}_i$$

$$= \sum_{j=1}^K \mathbb{E}\Big[ y_{ij}^2 - 2y_{ij}p_{ij} + p_{ij}^2 \Big] = \sum_{j=1}^K \Big( y_{ij}^2 - 2y_{ij}\mathbb{E}[p_{ij}] + \mathbb{E}[p_{ij}^2] \Big). \quad (5)$$

Among the three options presented above, we choose the last based on our empirical findings. We have observed the losses in Equations 3 and 4 to generate excessively high belief masses for classes and exhibit relatively less stable performance than Equation 5. We leave theoretical investigation of the disadvantages of these alternative options to future work, and instead, highlight some advantageous theoretical properties of the preferred loss below.

The first advantage of the loss in Equation 5 is that using the identity

$$\mathbb{E}[p_{ij}^2] = \mathbb{E}[p_{ij}]^2 + \mathrm{Var}(p_{ij}),$$

we get the following easily interpretable form

$$\mathcal{L}_i(\Theta) = \sum_{j=1}^K (y_{ij} - \mathbb{E}[p_{ij}])^2 + \mathrm{Var}(p_{ij})$$

$$= \sum_{j=1}^K \underbrace{(y_{ij} - \alpha_{ij}/S_i)^2}_{\mathcal{L}_{ij}^{err}} + \underbrace{\frac{\alpha_{ij}(S_i - \alpha_{ij})}{S_i^2(S_i + 1)}}_{\mathcal{L}_{ij}^{var}}$$

$$= \sum_{j=1}^K (y_{ij} - \hat{p}_{ij})^2 + \frac{\hat{p}_{ij}(1 - \hat{p}_{ij})}{(S_i + 1)}.$$

By decomposing the first and second moments, the loss aims to achieve the joint goals of minimizing the prediction error and the variance of the Dirichlet experiment generated by the neural net specifically for each sample in the training set. While doing so, it prioritizes data fit over variance estimation, as ensured by the proposition below.

**Proposition 1.** *For any $\alpha_{ij} \geq 1$, the inequality $\mathcal{L}_{ij}^{var} < \mathcal{L}_{ij}^{err}$ is satisfied.*

The next step towards capturing the behavior of Equation 5 is to investigate whether it has a tendency to fit to the data. We assure this property thanks to our next proposition.

**Proposition 2.** *For a given sample $i$ with the correct label $j$, $L_i^{err}$ decreases when new evidence is added to $\alpha_{ij}$ and increases when evidence is removed from $\alpha_{ij}$.*

A good data fit can be achieved by generating arbitrarily many evidences for all classes as long as the ground-truth class is assigned the majority of them. However, in order to perform proper uncertainty modeling, the model also needs to learn variances that reflect the nature of the observations. Therefore, it should generate more evidence when it is more sure of the outcome. In return, it should avoid generating evidences at all for observations it cannot explain. Our next proposition provides a guarantee for this preferable behavior pattern, which is known in the uncertainty modeling literature as *learned loss attenuation* [16].

**Proposition 3.** *For a given sample $i$ with the correct class label $j$, $L_i^{err}$ decreases when some evidence is removed from the biggest Dirichlet parameter $\alpha_{il}$ such that $l \neq j$.*

When put together, the above propositions indicate that the neural nets with the loss function in Equation 5 are optimized to generate more evidence for the correct class labels for each sample and helps neural nets to avoid misclassification by removing excessive misleading evidence. The loss also tends to shrink the variance of its predictions on the training set by increasing evidence, but only when the generated evidence leads to a better data fit. The proofs of all propositions are presented in the appendix.

The loss over a batch of training samples can be computed by summing the loss for each sample in the batch. During training, the model may discover patterns in the data and generate evidence for specific class labels based on these patterns to minimize the overall loss. For instance, the model may discover that the existence of a large circular pattern on MNIST images may lead to evidence for the digit zero. This means that the output for the digit zero, i.e., the evidence for class label $0$, should be increased when such a pattern is observed by the network on a sample. However, when counter examples are observed during training (e.g., a digit six with the same circular pattern), the parameters of the neural network should be tuned by back propagation to generate smaller amounts of evidence for this pattern and minimize the loss of these samples, as long as the overall loss also decreases. Unfortunately, when the number of counter-examples is limited, decreasing the magnitude of the generated evidence may increase the overall loss, even though it decreases the loss for the counter-examples. As a result, the neural network may generate some evidence for the incorrect labels. Such misleading evidence for a sample may not be a problem as long as it is correctly classified by the network, i.e., the evidence for the correct class label is higher than the evidence for other class labels. However, we prefer the total evidence to shrink to zero for a sample if it cannot be correctly classified. Let us note that a Dirichlet distribution with zero total evidence, i.e., $S = K$, corresponds to the uniform distribution and indicates total uncertainty, i.e., $u = 1$. We achieve this by incorporating a Kullback-Leibler (KL) divergence term into our loss function that regularizes our predictive distribution by penalizing those divergences from the *"I do not know"* state that do not contribute to data fit. The loss with this regularizing term reads

$$\mathcal{L}(\Theta) = \sum_{i=1}^{N} \mathcal{L}_i(\Theta) + \lambda_t \sum_{i=1}^{N} KL[D(\mathbf{p_i}|\tilde{\boldsymbol{\alpha}_i}) \,||\, D(\mathbf{p_i}|\langle 1,\ldots,1\rangle)],$$

where $\lambda_t = \min(1.0, t/10) \in [0, 1]$ is the annealing coefficient, $t$ is the index of the current training epoch, $D(\mathbf{p}_i|\langle 1,\ldots,1\rangle)$ is the uniform Dirichlet distribution, and lastly $\tilde{\boldsymbol{\alpha}}_i = \mathbf{y}_i + (1 - \mathbf{y}_i) \odot \boldsymbol{\alpha}_i$ is the Dirichlet parameters after removal of the non-misleading evidence from predicted parameters $\boldsymbol{\alpha}_i$ for sample $i$. The KL divergence term in the loss can be calculated as

$$KL[D(\mathbf{p}_i|\tilde{\boldsymbol{\alpha}}_i) \,||\, D(\mathbf{p}_i|\mathbf{1})]$$
$$= \log\left(\frac{\Gamma(\sum_{k=1}^{K}\tilde{\alpha}_{ik})}{\Gamma(K)\prod_{k=1}^{K}\Gamma(\tilde{\alpha}_{ik})}\right) + \sum_{k=1}^{K}(\tilde{\alpha}_{ik} - 1)\left[\psi(\tilde{\alpha}_{ik}) - \psi\left(\sum_{j=1}^{K}\tilde{\alpha}_{ij}\right)\right],$$

where $\mathbf{1}$ represents the parameter vector of $K$ ones, $\Gamma(\cdot)$ is the *gamma* function, and $\psi(\cdot)$ is the *digamma* function. By gradually increasing the effect of the KL divergence in the loss through the annealing coefficient, we allow the neural network to explore the parameter space and avoid premature convergence to the uniform distribution for the misclassified samples, which may be correctly classified in the future epochs.

## 5 Experiments

For the sake of commensurability, we evaluate our method following the experimental setup studied by Louizos et al. [24]. We use the standard LeNet with ReLU non-linearities as the neural network architecture. All experiments are implemented in Tensorflow [1] and the Adam [17] optimizer has been used with default settings for training.[1]

In this section, we compared the following approaches: (a) **L2** corresponds to the standard deterministic neural nets with softmax output and weight decay, (b) **Dropout** refers to the uncertainty estimation model used in [8], (c) **Deep Ensemble** refers to the model used in [21], (d) **FFG** refers to

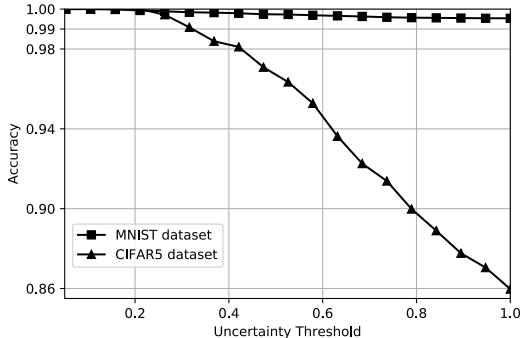

| Method | MNIST | CIFAR 5 |
|---|---|---|
| *L2* | 99.4 | 76 |
| *Dropout* | 99.5 | 84 |
| *Deep Ensemble* | 99.3 | 79 |
| *FFGU* | 99.1 | 78 |
| *FFLU* | 99.1 | 77 |
| *MNFG* | 99.3 | 84 |
| *EDL* | 99.3 | 83 |

Figure 2: The change of accuracy with respect to the uncertainty threshold for *EDL*.

Table 1: Test accuracies (%) for MNIST and CIFAR5 datasets.

the Bayesian neural net used in [18] with the additive parametrization [26], (e) **MNFG** refers to the structured variational inference method used in [24], (f) **EDL** is the method we propose.

We tested these approaches in terms of prediction uncertainty on MNIST and CIFAR10 datasets. We also compare their performance using adversarial examples generated using the Fast Gradient Sign method [10].

## 5.1 Predictive Uncertainty Performance

We trained the LeNet architecture for MNIST using $20$ and $50$ filters with size $5 \times 5$ at the first and second convolutional layers, and 500 hidden units for the fully connected layer. Other methods are also trained using the same architecture with the priors and posteriors described in [24]. The classification performance of each method for the MNIST test set can be seen in Table 1. The table indicates that our approach performs comparable to the competitors. Hence, our extensions for uncertainty estimation do not reduce the model capacity. Let us note that the table may be misleading for our approach, since the predictions that are totally uncertain (i.e., $u = 1.0$) are also considered as failures while calculating overall accuracy; such predictions with zero evidence implies that the model rejects to make a prediction (i.e. says *"I do not know"*). Figure 2 plots how the test accuracy changes if EDL rejects predictions above a varying uncertainty threshold. It is remarkable that the accuracy for predictions whose associated uncertainty is less than a threshold increases and becomes 1.0 as the uncertainty threshold decreases.

Our approach directly quantifies uncertainty using Equation 1. However, other approaches use entropy to measure the uncertainty of predictions as described in [24], i.e., uncertainty of a prediction is considered to increase as the entropy of the predicted probabilities increases. To be fair, we use the same metric for the evaluation of prediction uncertainty in the rest of the paper; we use Equation 2 for class probabilities.

In our first set of evaluations, we train the models on the MNIST train split using the same LeNet architecture and test on the notMNIST dataset, which contains letters, not digits. Hence, we expect predictions with maximum entropy (i.e. uncertainty). On the left panel of Figure 3, we show the empirical CDFs over the range of possible entropies $[0, \log(10)]$ for all models trained with MNIST dataset. The curves closer to the bottom right corner of the plot are desirable, which indicate maximum entropy in all predictions [24]. It is clear that the uncertainty estimates of our model is significantly better than those of the baseline methods.

We have also studied the setup suggested in [24], which uses a subset of the classes in CIFAR10 for training and the rest for out-of-distribution uncertainty testing. For fair comparison, we follow the authors and use the large LeNet version which contains 192 filters at each convolutional layer and has 1000 hidden units for the fully connected layers. For training, we use the samples from the first five categories {dog, frog, horse, ship, truck} in the training set of CIFAR10. The accuracies of the trained models on the test samples from same categories are shown in Table 1. Figure 2 shows that *EDL* provides much more accurate predictions as the prediction uncertainty decreases.

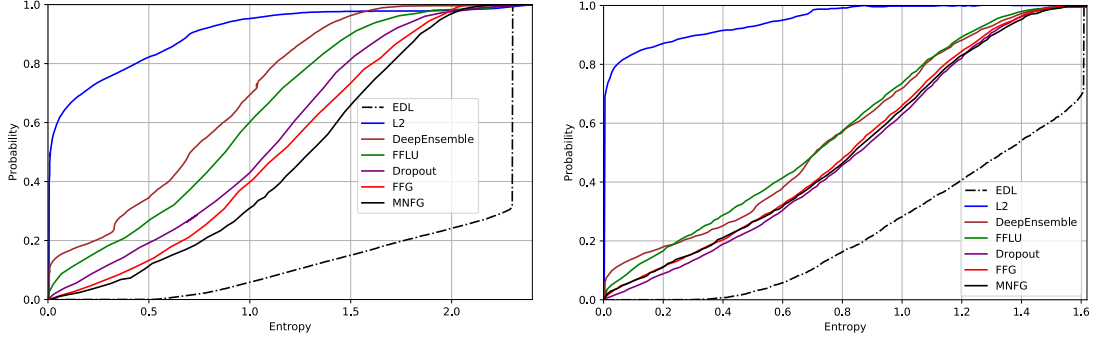

Figure 3: Empirical CDF for the entropy of the predictive distributions on the notMNIST dataset (left) and samples from the last five categories of CIFAR10 dataset (right).

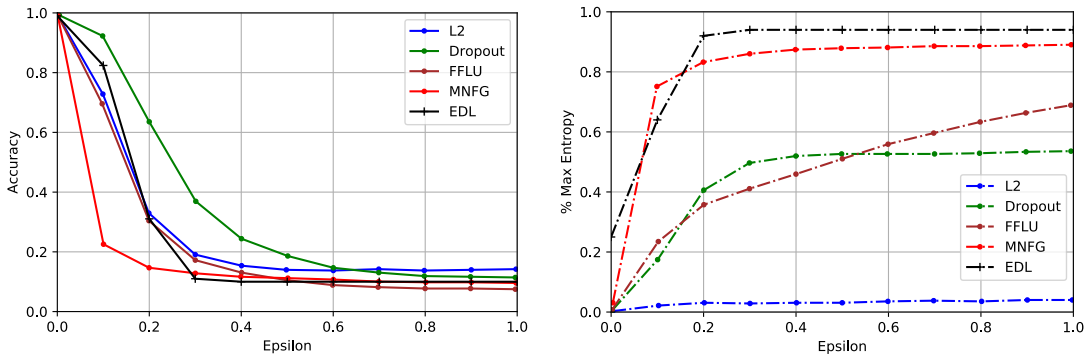

Figure 4: Accuracy and entropy as a function of the adversarial perturbation $\epsilon$ on the MNIST dataset.

To evaluate the prediction uncertainties of the models, we tested them on the samples from the last five categories of the CIFAR10 dataset, i.e., {airplane, automobile, bird, cat, deer}. Hence, none of the predictions for these samples is correct and we expect high uncertainty for the predictions. Our results are shown at the right of Figure 3. The figure indicates that *EDL* associates much more uncertainty to its predictions than other methods.

## 5.2 Accuracy and Uncertainty on Adversarial Examples

We also evaluated our approach against adversarial examples [10]. For each model trained in the previous experiments, adversarial examples are generated using the Fast Gradient Sign method from the Cleverhans adversarial machine learning library [28], using various values of adversarial perturbation coefficient $\epsilon$. These examples are generated using the weights of the models and it gets harder to make correct predictions for the models as the value of $\epsilon$ increases. We use the adversarial examples to test the trained models. However, the *Deep Ensemble* model is excluded in this set of experiments for fairness, since it is trained on adversarial examples.

Figure 4 shows the results for the models trained on the MNIST dataset. It demonstrates accuracies on the left panel and uncertainty estimations on the right. Uncertainty is estimated in terms of the ratio of prediction entropy to the maximum entropy, which is referred to as % max entropy in the figure. Let us note that the maximum entropy is $\log(10)$ and $\log(5)$ for the MNIST and CIFAR5 datasets, respectively. The figure indicates that Dropout has the highest accuracy for the adversarial examples as shown on the left panel of the figure; however, it is overconfident on all of its predictions as indicated by the right figure. That is, it places high confidence on its wrong predictions. However, *EDL* represents a good balance between the prediction uncertainty and accuracy. It associates very high uncertainty to the wrong predictions. We perform the same experiment on the CIFAR5 dataset. Figure 5 demonstrates the results, which indicates that *EDL* associates higher uncertainty for the wrong predictions. On the other hand, other models are overconfident with their wrong predictions.

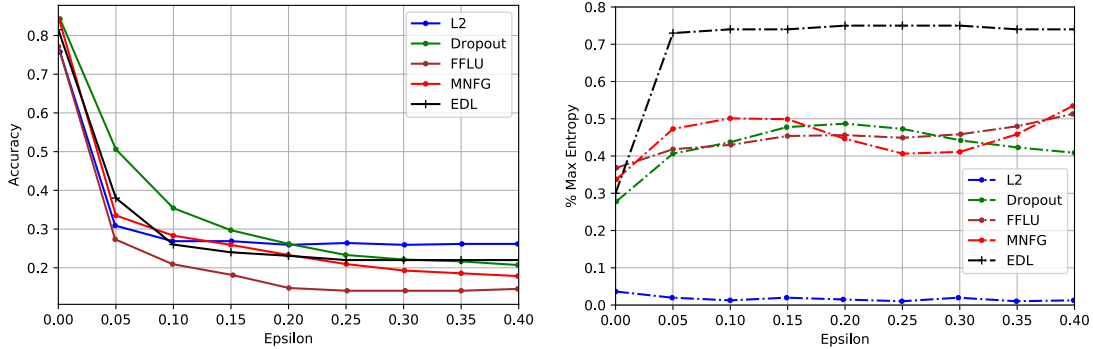

Figure 5: Accuracy and entropy as a function of the adversarial perturbation $\epsilon$ on CIFAR5 dataset.

## 6 Related Work

The history of learning uncertainty-aware predictors is concurrent with the advent of modern Bayesian approaches to machine learning. A major branch along this line is Gaussian Processes (GPs) [29], which are powerful in both making accurate predictions and providing reliable measures for the uncertainty of their predictions. Their power in prediction has been demonstrated in different contexts such as transfer learning [15] and deep learning [32]. The value of their uncertainty calculation has set the state of the art in active learning [12]. As GPs are non-parametric models, they do not have a notion of deterministic or stochastic model parameters. A significant advantage of GPs in uncertainty modeling is that the variance of their predictions can be calculated in closed form, although they are capable of fitting a wide spectrum of non-linear prediction functions to data. Hence they are universal predictors [31].

Another line of research in prediction uncertainty modeling is to employ prior distributions on model parameters (when the models are parametric), infer the posterior distribution, and account for uncertainty using high-order moments of the resultant posterior predictive distribution. BNNs also fall into this category [25]. BNNs build on accounting for parameter uncertainty by applying a prior distribution on synaptic connection weights. Due to the non-linear activations between consecutive layers, calculation of the resultant posterior on the weights is intractable. Improvement of the approximation techniques, such as Variational Bayes (VB) [2, 6, 27, 23, 9] and Stochastic Gradient Hamiltonian Monte Carlo (SG-HMC) [3] tailored specifically for scalable inference of BNNs is an active research field. Despite their enormous prediction power, the posterior predictive distributions of BNNs cannot be calculated in closed form. The state of the art is to approximate the posterior predictive density with Monte Carlo integration, which brings a significant noise factor on uncertainty estimates. Orthogonal to this approach, we bypass inferring sources of uncertainty on the predictor and directly model a Dirichlet posterior by learning its hyperparameters from data via a deterministic neural net.

## 7 Conclusions

In this work, we design a predictive distribution for classification by placing a Dirichlet distribution on the class probabilities and assigning neural network outputs to its parameters. We fit this predictive distribution to data by minimizing the Bayes risk with respect to the L2-Norm loss which is regularized by an information-theoretic complexity term. The resultant predictor is a Dirichlet distribution on class probabilities, which provides a more detailed uncertainty model than the point estimate of the standard softmax-output deep nets. We interpret the behavior of this predictor from an evidential reasoning perspective by building the link from its predictions to the belief mass and uncertainty decomposition of the subjective logic. Our predictor improves the state of the art significantly in two uncertainty modeling benchmarks: i) detection of out-of-distribution queries, and ii) endurance against adversarial perturbations.

**Acknowledgments**

This research was sponsored by the U.S. Army Research Laboratory and the U.K. Ministry of Defence under Agreement Number W911NF-16-3-0001. The views and conclusions contained in this document are those of the authors and should not be interpreted as representing the official policies, either expressed or implied, of the U.S. Army Research Laboratory, the U.S. Government, the U.K. Ministry of Defence or the U.K. Government. The U.S. and U.K. Governments are authorized to reproduce and distribute reprints for Government purposes notwithstanding any copyright notation hereon. Also, Dr. Sensoy thanks to the U.S. Army Research Laboratory for its support under grant W911NF-16-2-0173.

## Footnotes

[1]The implementation and a demo application of our model is available under https://muratsensoy.github.io/uncertainty.html

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
