[Supplementary Material]

## Appendix: Propositions for the Loss Function in (5)

**Proposition 1.** For any $\alpha_{ij} \geq 1$, the inequality $\mathcal{L}_{ij}^{var} < \mathcal{L}_{ij}^{err}$ is satisfied.

**Proof.** Consider $y_j = 0$, then $\mathcal{L}_{ij}^{err} = \alpha_{ij}^2/S_i^2$. As $\dfrac{(S_i - \alpha_{ij})}{(S_i + 1)} < 1$ and $\dfrac{\alpha_{ij}}{S_i^2} \leq \dfrac{\alpha_{ij}^2}{S_i^2}$ we get

$$\frac{\alpha_{ij}(S_i - \alpha_{ij})}{S_i^2(S_i + 1)} < \frac{\alpha_{ij}^2}{S_i^2}.$$

Now consider $y_j = 1$, then

$$\mathcal{L}_{ij}^{err} = \left(1 - \frac{\alpha_{ij}}{S_i}\right)^2 = \frac{(S_i - \alpha_{ij})^2}{S_i^2}.$$

As $(S_i - \alpha_{ij}) > \alpha_{ij}/(S_i + 1)$, we attain

$$\frac{\alpha_{ij}(S_i - \alpha_{ij})}{S_i^2(S_i + 1)} < \frac{(S_i - \alpha_{ij})^2}{S_i^2} \quad \blacksquare$$

**Proposition 2.** For a given sample $i$ with the correct label $j$, $L_i^{err}$ decreases when new evidence is added to $\alpha_{ij}$ and increases when evidence is removed from $\alpha_{ij}$.

**Proof.** Let $\nu$ represent evidence to be added to the Dirichlet parameter $\alpha_{ij}$. Then, $L_i^{err}$ is updated as

$$\hat{L}_i^{err} = \left(1 - \frac{\alpha_{ij} + \nu}{S_i + \nu}\right)^2 + \sum_{k \neq j}\left(\frac{\alpha_{ik}}{S_i + \nu}\right)^2$$

which is smaller than $L_i^{err}$ for $\nu > 0$, since

$$\left(1 - \frac{\alpha_{ij} + \nu}{S_i + \nu}\right)^2 < \left(1 - \frac{\alpha_{ij}}{S_i}\right)^2 \quad \text{and}$$

$$\sum_{k \neq j}\left(\frac{\alpha_{ik}}{S_i + \nu}\right)^2 < \sum_{k \neq j}\left(\frac{\alpha_{ik}}{S_i}\right)^2$$

Similarly $\hat{L}_i^{err}$ becomes greater than $L_i^{err}$ when $\nu < 0$ $\blacksquare$

**Proposition 3.** For a given sample $i$ with the correct class label $j$, $L_i^{err}$ decreases when some evidence is removed from the biggest Dirichlet parameter $\alpha_{il}$ such that $l \neq j$.

**Proof.** Let the expected value of the predicted Dirichlet distribution for the sample $i$ be $[\hat{p}_{i1}, \ldots, \hat{p}_{iK}]$. When some evidence is removed from $\alpha_{il}$, $\hat{p}_{il}$ decreases by $\delta_{il} > 0$. As a result, $\hat{p}_{ik}$ for all $k \neq l$ increases by $\delta_{ik} > 0$ such that $\sum_{k \neq l} \delta_{ik} = \delta_{il}$, since the expected values must sum to one (2). Let $\tilde{p}_{il}$ be the updated expected value for the $l^{th}$ component of the Dirichlet distribution after the removal of evidence. Then, $L_i^{err}$ before the removal of evidence can be written as

$$L_i^{err} = (1 - \hat{p}_{ij})^2 + \left(\tilde{p}_{il} + \sum_{k \neq l} \delta_{ik}\right)^2 + \sum_{k \notin \{j,l\}} \hat{p}_{ik}^2$$

and it is updated after the removal of evidence as

$$\tilde{L}_i^{err} = (1 - \hat{p}_{ij} - \delta_{ij})^2 + \tilde{p}_{ij}^2 + \sum_{k \notin \{j,l\}} (\hat{p}_{ik} + \delta_{ik})^2$$

the difference $L_i^{err} - \tilde{L}_i^{err}$ becomes

$$\underbrace{2(1 - \hat{p}_{ij})\delta_{ij}}_{\geq 0} + 2\left(\tilde{p}_{il} \sum_{k \neq l} \delta_{ik} - \sum_{k \notin \{j,l\}} \hat{p}_{ik}\delta_{ik}\right) + \underbrace{\left(\left(\sum_{k \neq l} \delta_{ik}\right)^2 - \sum_{k \neq l} \delta_{ik}^2\right)}_{\geq 0},$$

which is always positive for $\hat{p}_{il} > \tilde{p}_{il} \geq \hat{p}_{ik}$ (s.t. $k \neq j$) and maximizes as $\hat{p}_{il}$ increases $\blacksquare$