[Reviews · NeurIPS 2018]

Reviewer 1



The authors propose a method to quantify prediction uncertainty by placing a Dirichlet prior on the softmax output of a neural net. The idea is to let the algorithm say "I don't know" if not enough evidence is provided for any of the classes. The optimization of the new loss function improves uncertainty estimation while keeping an accuracy inline with other approaches when compared on benchmark datasets. They also show the value for the detection of out-of-samples queries or for robustness towards adversarial setting. The quality of the work is really good, with interesting modeling ideas which builds upon the Dempster–Shafer Theory of Evidence (DST) and subjective logic. Theoretical proofs (in short appendix) guarantee the properties of the approach. The authors also compare their algorithm on benchmark datasets to numerically illustrate the performance of the approach. In particular, the algorithm has on-par accuracy as the other models (from recent publications) while being better able to quantify uncertainty and being robust to adversarial settings. The paper is clearly written and easy to follow. It is well motivated and addresses an important limitation of a lot of classification algorithms, which is novel as far as I know.

Reviewer 2



Summary This paper leverages the theory of evidence to develop a framework for characterizing uncertainty in deep learnig and then evaluates the results on two standard benchmark datasets relative to a wide selection of benchmark methods. . The paper poses a probabilistic framework on the outputs of the neural net in order to develop a means of modeling the uncertainty. Strengths: Quantifying uncertainty is an important direction and an oft-cited weakness of deep learning adn a reason that deep learning is in appropriate for many problems solutions like the provied will enable extension of the impressive accuracies of deep learning into domaiss it has not yet been sucessful or trusted to date. The experimental results are impressive, in particular the robustness to adversarial examples. The authors use the standard data sets in nonstandard cmbinations of tst train splits in order to deelop an appropriate framework Weaknesses: figure 1 caption, degrees should be angle line 41: why call it simple? this should be qualified or the word ommitted. (minor) inconsistent refeence to left vs right subfigured (eg compare figures 1 and 3)

Reviewer 3



- Novelty is below the NIPS level. The claim of using the DST theory was stated in the beginning (L36) as: achieving a "richer" representation. I did not see through the rest of the paper why this is a "richer" representation. - The experiments are rigorously performed and analysed. They well serve the cause of the paper. - The text in the introduction is trying to be too verbose for a scientific paper at this level. There is no need for that in my opinion. An example is: "All these (no longer) futuristic ... Are they futuristic or are they not? I'd go in favour of a more rigorous and unequivocal (and still eye-catching) presentation style. - L85: 13]. "Hence, it allows one to use the principles of evidential theory to quantify belief masses and uncertainty through a well-defined theoretical framework.": Ok, but why is using the principles of evidential theory preferred? A Dirichlet can be used without resorting to this theory, i.e. in a Bayesian setting. The motivation can be a bit clearer here too (in the beginning of Section 3, I mean). - There are some grammar mistakes and typos. Examples include: -- L64: "The consequence is then unreliability of ..." -- L75: "our approach proposed in this paper ..." - L22: Who said that the "saturation" is "fast approaching"? I mean are there citations demonstrating that? Or is that just a speculation?